# Simulation of Solvatochromic Phenomena in Xanthione Using Explicit Solvent Methods

**DOI:** 10.3390/molecules29235609

**Published:** 2024-11-27

**Authors:** Anjay Manian, Zifei Chen, Rohan J. Hudson, Salvy P. Russo

**Affiliations:** 1ARC Centre of Excellence in Exciton Science, School of Science, RMIT University, Melbourne 3000, Australia; zifei.chen@rmit.edu.au; 2ARC Centre of Excellence in Exciton Science, School of Chemistry, University of Melbourne, Parkville 3010, Australia; rohan.hudson@moglabs.com; 3MOGLabs, Carlton 3053, Australia

**Keywords:** xanthione, solvent effects, explicit solvation, DFT, MRCI, MD

## Abstract

Xanthione is a sulfated polycyclic aromatic hydrocarbon which exhibits unique anti-Kasha properties and substantial sensitivity to its medium. Due to this sensitivity however, this makes xanthione-based systems very difficult to simulate. Further, xanthione’s is understood to be come more photostable in the presence of a highly polar medium, however whether these photophysical properties could be taken advantage of for certain applications remains to be seen. In clarifying long-held beliefs of specific solvent effects, we apply a rigorous theoretical solvent analysis in both implicit and explicit solvent mediums to elucidate a more complete description of solvent polarity sensitivity in xanthione using both quantum chemical and molecular dynamics techniques. Not only was it found that explicit solvation methods are vital in an accurate description of the system, only a handful of explicit solvent molecules in the simulation are required to yield an appropriate electronic description. This short work is vital to devising future applications for xanthione-based and other quantum technologies, and is an important foundation stone on this journey.

## 1. Introduction

Xanthones are polycyclic aromatic hydrocarbon which have been shown to display tuneable photophysical properties with respect to their atomic surroundings [1,2,3]. Indeed, a shift in photoluminescence quantum yield from 0.0002 in cyclohexane to 0.46 in trifluoroethanol is observable [4], highlighting a distinct solvatochromic nature. The 1980s held the most interest for sulfated xanthones, or xanthiones, however interest eventually waned after it was concluded that there were some solvent-specific effects influencing excited-state properties. However, more recently interest has begun to increase again due to their anti-Kasha optical properties and suspected singlet fission qualities [5], and is also solvatochromic in nature [6,7], with new studies being reported regularly. However, experimental testing is both time consuming and financially expensive; therefore a theoretical treatment may be more prudent.

However, accurate theoretical modelling of solvatochromic systems is often difficult if not impossible; due primarily to implicit solvent models like the polarisable continuum model (PCM), whereby the solvent is approximated using a dielectric cavity, not capturing the important electron structure properties that comes with solute-solvent bonding interactions. This phenomenon is observable in a myriad of systems, such as in indole [8,9], 2.4-bismidazolylphenol [10], and naphthalene diimide [11,12].

There is also the question of how complete the explicit solvation need be. Our previous work on indole [8] highlighted that there is typically some saturation limit on the number of solvent molecules that need to be included in the simulation. If we take a given xanthione chromophore, it is not clear how many solvent molecules are required to replicate experimental conditions. In addition, singlet fission is highly sensitive to the energy of the low-lying excited states [13]; if we consider that both the optically dark and bright states can be influenced by the polarity of a given solvent, can we use this information to fuel future xanthione-singlet fission studies?

This work will seek to gain further clarity concerning these important questions. This short but vital work will use 3,6-bis(diethylamino)-xanthene-9-thione as our xanthione derivative (Figure 1) to elucidate how solvent polarity effects the energies of both the optically dark and bright states, referred hereafter to as the $L_{a}$ and $L_{b}$ state, respectively (similar to Platt’s notation [14]). Vertical excitation energies at both Franck-Condon points as well as the electronic structure will be calculated within implicit and explicit solvent models, in order to further understand how solvent polarity alters the photophysical properties of xanthione.

## 2. Computational Details

Geometries for the electronic ground $S_{0}$ state, and the first and second electronic excited $L_{a}$ and $L_{b}$ states, were all calculated using the Coulomb-attenuated Becke 3-parameter Lee–Yang–Parr (CAM-B3LYP) [15,16,17,18,19] exchange-correlation functional with the damped Becke-Johnson geometry dependent 3-parameter DFT-D3(BJ) dispersion correction [20,21], alongside the double-ζ 6-primitive Slater-type orbital 3-inner and 1-outer 6–31G basis set [22,23,24], as implemented in the Gaussian 16 software package [25]. Implicit solvent systems (xanthione only) were simulated using a PCM via a ground state approach. Here, the self-consistent field is corrected through a solvent-effects term, allowing for optimisation within solution of excited states. Explicit solvent models were optimised by placing the xanthione alongside explicit solvent molecules, in addition to a PCM. Explicit solvation was performed in sets of 2, 4, and 6, i.e., xanthione and n-solvent molecules within a PCM.

Initial geometries were chosen based on a simplified charge analysis. Following optimisation of the gas phase geometries, 2 solvent molecules were injected into the system above and below the xanthione plane favouring the sulfur-site. Following successful optimisation, 2 more were injected above and below the oxygen-site, and following optimisation 2 more on either side near the nitrogen-centres. This was done for each solute-solvent system.

To investigate the effect of solvent polarity on the excited-state manifold of xanthione, solvents were selected primarily based on their dielectric constant, rather than their solubility with respect to xanthione. While solubility is typically a key factor in solvent selection for such studies, the focus on dielectric constant allowed for a broader exploration of solvent polarity effects. It should be noted, however, that some of the solvents used in this work, though less commonly employed with xanthione due to solubility limitations, still provide valuable insights into the solute-solvent interaction of xanthione. The solvents used in this work are listed in Table 1.

Single-point calculations for each optimised structure was performed using the DFT-based multireference configuration interaction DFT/MRCI method [26,27]. The one-particle basis was calculated using the the Becke “half-and-half” Lee-Yang-Par BHLYP functional [28,29] with the DFT-D3(BJ) correction and the Karlsruhe variant of the split valence with polarisation functions on non-hydrogen atoms def2-SV(P) basis set [30,31], as implemented in the Turbomole software package [32]. The DFT/MRCI reference space was generated iteratively by including all electron configurations with expansion coefficients greater than $10^{-3}$ in numerous probe calculations, using 10 electrons across 10 orbitals, allowing only for a maximum of two-electron excitations. Probe runs were calculated by discarding configurations with energy less than the highest reference energy; starting with a threshold of 0.6 $E_{h}$, then 0.8, with the finalised wavefunction built using a threshold of 1.0. Molecular orbitals with energies larger than 2.0 $E_{h}$ were not used. We note that n = 4, 6 in toluene was not resolved due to computational limitations.

To further demonstrate the impact the chosen number of solvent molecules has in explicit solvent models, we also perform molecular dynamics (MD) simulations. Here, we have chosen four typical solvents: toluene, acetone, dimethyl sulfoxide (DMSO), and water. The simulation system included one xanthione molecule and 400 solvent molecules (800 in the case of water). Periodic boundary conditions are employed. First, a 5 ns isobaric-isothermal (NPT) ensemble simulation was conducted to equilibrate the system at room temperature, yielding stable boxes with lengths 42.5 Å, 37.6 Å, 37.2 Å, and 29.2 Å, for each of the 4 solvents respectively. This was followed by a 5 ns canonical (NVT) ensemble simulation to sample the system for statistical analysis. The MD simulations were carried out using the GROMACS software package [33], employing the General Amber Force Field (GAFF) [34] for both solute and solvent. Effective charges were calculated using the restrained electrostatic potential (RESP) fitting method [35,36], using the B3LYP density functional and the Karlsruhe variant of the triple-ζ valence polarised def2-TZVP basis set [31,37]. Electrostatic interactions were treated using the particle-mesh Ewald (PME) method [38], while Lennard-Jones interactions were cut off at 8 Å. Temperature control was maintained via the Nosé–Hoover thermostat [39,40], and pressure was regulated in the NPT simulations using the Berendsen barostat [41].

## 3. Results & Discussion

For xanthione, a dominant leading configuration state function (CSF) of one below the highest occupied molecular orbital (HOMO-1) to the lowest unoccupied molecular orbital (LUMO) is observed for the $L_{a}$ state, while for the $L_{b}$ state a dominant HOMO→LUMO CSF is noted. In the gas phase, there is some level inversion between the non-bonding $n\pi^{★}$ state and a nearby $\sigma \pi^{★}$ state, however upon solvation (independent of polarity) the $L_{a}$ state is of $n\pi^{★}$ character. In all cases, the $L_{b}$ state is bright, and is of $\pi \pi^{★}$ character. These state assignments agree well with reported chemistry [42] in 3-methylpentane (ϵ = 1.895 [43]).

Examination of the fluorescence qualities of xanthione in various solvents shows that for the most part PCMs tend to overestimate vertical excitation energies with respect to the trends observed from explicit solvation. The absorption energies (Figure 2) of the $L_{b}$ state show implicit solvent energies to trend upwards from around 2.35 eV and plateaus at ∼2.57 eV. X = 2 explicit solvation trends lower in energy by more than 0.1 eV, with larger solvent shells of similar quality, but progressively lower. Outliars to this trend are observable for X = 2, 4 in dichloroethane, X = 2 in acetone, and X = 6 in dimethyl sulfoxide. Like the $L_{b}$ states, PCM energies are overestimated by 0.8 eV for the $L_{a}$ manifold, but no large outliars are observable in the data, with a relatively smooth progression in energies. The emission profiles are significantly more variable compared to the absorption spectra; while implicit solvents models yield overestimated energies as with the absorption spectra, the difference is larger: around 0.15 eV on the $L_{b}$ manifold, and up to 0.6 eV on the $L_{a}$ manifold. Interesting, most of the data points in the lower solvent polarity region appear to not follow any clear trend. We suspect that this is due to solvent drift (since the solvent is not frozen between optimisations, small changes may have large impacts on the electronic structure, and therefore the energies), however of note is a large disagreement for X = 4, 6 in dimethyl sulfoxide on both $L_{a}$ and $L_{b}$ manifolds.

It is very important to note that for many of these systems, direct absorption to the $L_{a}$ manifold is more energetically favourable than excitation to the $L_{b}$ state. in other words, there is no adiabatic level inversion between the $L_{a}$ and $L_{b}$ state in xanthione noted in this work (Figure 3), however there is a large reorganisation energy in some of these compounds, likely due again to solvent drift.

A closer inspection of the electronic structure (see ESI) shows that while implicit solvents show identical electron densities for both $L_{a}$ and $L_{b}$ states, explicit solvation results in some exciton delocalisation effects through solute-solvent interactions for the $L_{a}$ state specifically. Indeed, for X = 2 in toluene for example, a small portion of the density can be observed shunted across to the solvent molecules. This is slightly more pronounced in X = 2 in acetone, but is most prevalent in X = 6 dimethyl sulfoxide. This behaviour does not appear in either the $n\pi^{★}$ or $\pi \pi^{★}$ manifolds. If we note the systems studied in this work as snapshots of an aggregate system studied in experiment, we can begin to understand why some data points in Figure 2 appear erroneous; these geometries can be considered as lower or upper bounds of an aggregate energetic profile.

Despite these complications, these results correlate fairly well with experiment. Assuming $L_{b}$ emission, Bondarev and co-workers [2] for example, for unsubstituted xanthione in n-hexane (ϵ = 1.882) report absorption and emission energies of 3.08 eV and 2.76 eV, respectively (2.73 eV and 2.33 eV in this work for toluene), and in acetonitrile energies of 3.05 eV and 2.73 eV, respectively (2.71 eV and 2.56 eV in this work). Lorenc and co-workers [44] reported similar energies in acetonitrile, with absorption and emission energies of 3.02 eV and 2.73 eV, respectively. Maciejewski & Steer [45] report the absorption energy to be between 2.97–3.65 eV in perfluoro-n-hexane (ϵ = 1.76 [46]) depending on the derivative, and a $S_{2}\to S_{1}$ energy gap between 0.99–1.36 eV (0.40 ev in this work for toluene). This agrees with Capitanio and co-workers [42] who reported an absorption energy of 2.93 eV in 3-methylpentane (ϵ = 1.90 (from the Hazardous Substances Data Bank)). Maciejewski and co-workers [47] also report on absorption and emission characteristics in a myriad of different mediums; absorption energies of 3.16 eV in perfluoro-n-hexane, 2.91 eV in water (2.70 eV in this work), and 2.98 eV in micellar sodium perfluorooctanoate, and emission energies of 2.76 eV in perfluoro-n-hexane, and 2.67 eV in micellar sodium perfluorooctanoate (we are unable to source a dielectric constant for this medium, however as it is a micellar system we suspect aggregation to play an important role). There is one notable disagreement with literature, related to the $n\pi^{★}$ state: Capitanio and co-workers [42] also report a 3-methylpentane $L_{b}$ absorption energy of 1.96 eV, which is 0.3 eV lower than any trends observed in this work. We suspect this is due to aggregation-induced energy level splitting of xanthione in solution.

From these results, we can see that concerning how much computational effort is required for explicit solvent methods, X = 2, 4 is typically enough to yield a good return on accuracy with respect to computational cost; X = 6 is just not necessary as the quality in results between X = 4 compared to X = 6 is within 10% of each other. Importantly, one needs to question whether the additional computational cost is worth that additional resolution. This data suggests that energies are most strongly influenced by how the solvent interacts with the xanthione core. Specifically, in the case of the highly polar solvents, hydrogen bonding can be observed to be commonplace. Accordingly, this behaviour is not captured by implicit models, but is by explicit models of X = 2, 4. However and as already mentioned, the configurations studied in this work as snapshots of a dynamic aggregate, and may not be representative of the molecular ensemble. In terms of some ensemble average, it is difficult to know for certain how the configurations studied in this work relate to this ensemble average, and is a very common problem when dealing with solvent sensitive systems [8]. This is what we mean when we say “outliars can be considered upper/lower bounds”; a configuration in the centre of the ensemble average would be expected to obey the general trends, however those configurations found in this study do not. Therefore, they likely fall within the upper or lower limits of the ensemble average. To further investigate the reasonable nature of our initial conditions to optimisation (as this may effect the final result), the bulk configuration can be probed using molecular dynamics. Analysis of the center-of-mass radial distribution function g(r) for solvent molecules relative to the xanthione solute (Figure 4a) shows the first solvation shell to manifest at around 2 Å, while beyond 8 Å, the system behaves like a bulk solution (g(r)=1). The region between 2–8 Å acts as a buffer, or transition layer. Notably, at 4 Å, g(r) deviates from the bulk value by less than 10%, indicating that the solvation process quickly approaches bulk behavior. At this distance, the number of nearest neighbour solvent molecules is approximately four (Figure 4b), true for acetone, toluene and DMSO. For water, a slower shift in the transition layer is found for the radial distribution function. However, the use of six water molecules remains a reasonable approximation for the solvation shell, correlating with previous observations of water in explicit solvent models [8]. This suggests that even with a relatively small number of solvent molecules, a realistic solvation process can be captured.

Interestingly, while g(r) and the first solvation shell exhibit similar behaviour for all chosen solvents, their effects on xanthione differ significantly. The root mean square deviation (RMSD) analysis (Figure 4c) reveals two distinct peaks: the first corresponding to the implicit solvent geometry, and the second representing solvent-induced distortions. Among the solvents studied, toluene exhibited the largest deviation from its implicit solvent structure. As the dielectric constant of the solvent increases, RMSD decreases, indicating that solvent configurations deviate more from their gas-phase geometry in low-polarity environments. In contrast, solvents with higher dielectric constants lead to geometries more closely resembling the gas-phase structure. This suggests that low-polarity solvents may not fully sample the configuration space, instead favouring a limited set of atomic arrangements, which could introduce fluctuations in the results of quantum chemistry calculations if the implicit solvent model’s configurations is close to its corresponding geometry. To address this, more accurate statistical ab initio methods should be incorporated in future work to improve the reliability of these simulations.

Examination of the radial distribution function of the heteroatoms (Figure 4d) shows that no strong interaction is observed between xanthione and DMSO. However, a strong solute-solvent interaction breaks the symmetry of the xanthione core and perturbs the equilibrium geometry compared to the symmetrical core observed in an implicit solvents; this interaction serves as a possible explanation for the explicit solvent results (Figure 3). No clear difference in the radial distribution function between acetone and DMSO is observable.

Further on the solvents stabilisation; using an explicit model one would typically expect more of a pronounced effect between the solvent molecule and the sulfur-centre. However, while this work does capture the subtle environmental differences near the oxygen and nitrogen sites, it does not for the sulfur-centres in terms of the molecular configurations. Examination of the charge sites (−0.39 au, −0.35 au, and −0.41 au for S, O, and N, respectively) suggests that the contribution of the sulfur to overall dipole moment is not distinctive. That being said, even with similar solvent configurations it will likely still influence the electronic structure. Therefore, we acknowledge that while we observe no specific contribution due to the sulfur-centre, the electronic structure shows a clear dependence on explicit solvent interactions at this site.

## 4. Conclusions

This study provides a comprehensive investigation into the solvent effects on the xanthione chromophore, utilising both implicit and explicit solvent models. The use of implicit solvation via a PCM was found to significantly overestimate vertical excitation energies; this discrepancy was effectively corrected by incorporating explicit solvent molecules. While no inversion of the $L_{a}$ and $L_{b}$ states was detected, the $L_{a}$ state consistently emerged as the lowest energy manifold in highly polar solvents. Molecular dynamics simulations further revealed that capturing the essential solvent-induced effects requires only a small number of explicit solvent molecules, with convergence achieved using 4–6 molecules for bulk properties, with less than a 10% error margin. Importantly, these results infer that energy levels as a function of solvent polarity trend towards a plateau once solvent polarity is beyond 60; while the $L_{b}$ state is largely unaffected by this, the $L_{a}$ state is shown to be highly sensitive when examining the adiabatic energy, while the $L_{b}$ state is more sensitive in terms of vertical excitation energies. The energy gap is shown to be minimal for higher polarity mediums than lower.

This work lays a solid foundation for further exploration into the complex solvatochromic behaviour of xanthione, offering key insights into how solvent polarity influences its photophysical stability. The findings suggest that polar solvents can significantly stabilise xanthione’s electronic states as well as its implicit solvent geometry, but also underscores the potential for further refinement in our understanding of solvent-induced modifications to its electronic structure. These results open avenues for future development of more precise quantum chemical models, which will be essential in harnessing xanthione’s unique properties for applications in emerging quantum technologies.

## Figures and Tables

**Figure 1 molecules-29-05609-f001:**
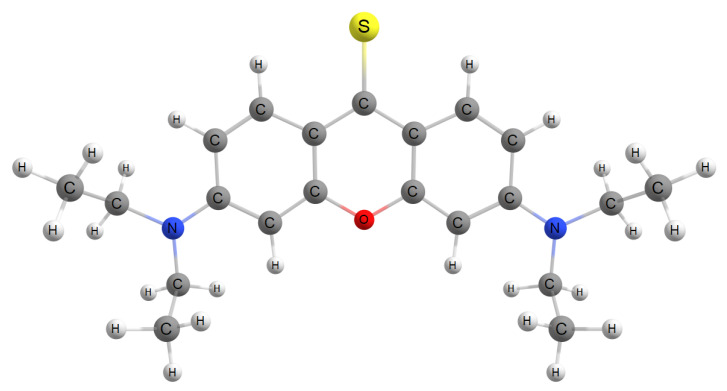
The xanthione compound used in this work. Its detailed name is 3,6-bis(diethylamino)-xanthene-9-thione.

**Figure 2 molecules-29-05609-f002:**
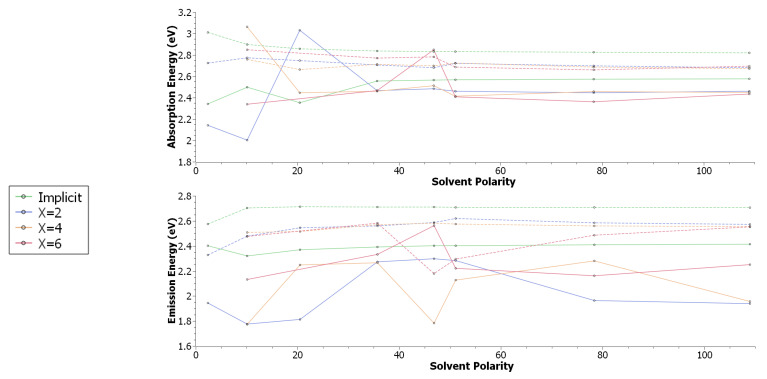
Xanthione vertical absorption and emission energies (DFT/MRCI) calculated for a polarisable continuum model and with X number of explicit solvent molecules in addition to a polarisable continuum model, as a function of the solvent polarity. Solid lines refer to the $L_{a}$ state, while dashed lines refer to the $L_{b}$ state.

**Figure 3 molecules-29-05609-f003:**
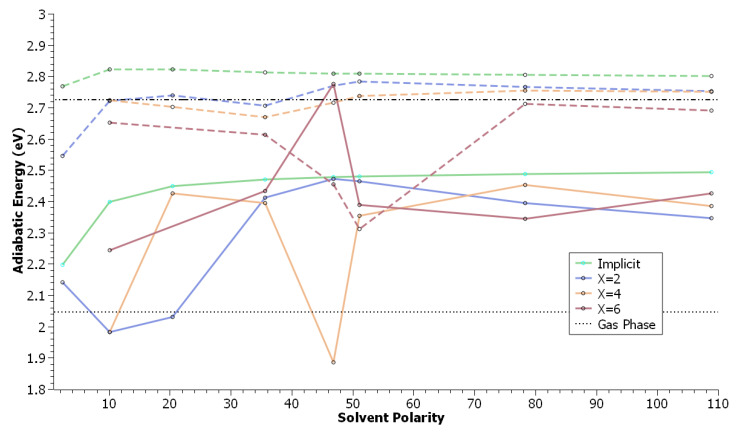
Calculated adiabatic energies (DFT/MRCI) for Xanthione in a polarisable continuum model and with X number of explicit solvent molecules in addition to a polarisable continuum model, as a function of the solvent polarity. Solid lines refer to the $L_{a}$ state, while dashed lines refer to the $L_{b}$ state. Black lines show gas phase energies for the $L_{a}$ (dashed) and $L_{b}$ (dot-dashed) states.

**Figure 4 molecules-29-05609-f004:**
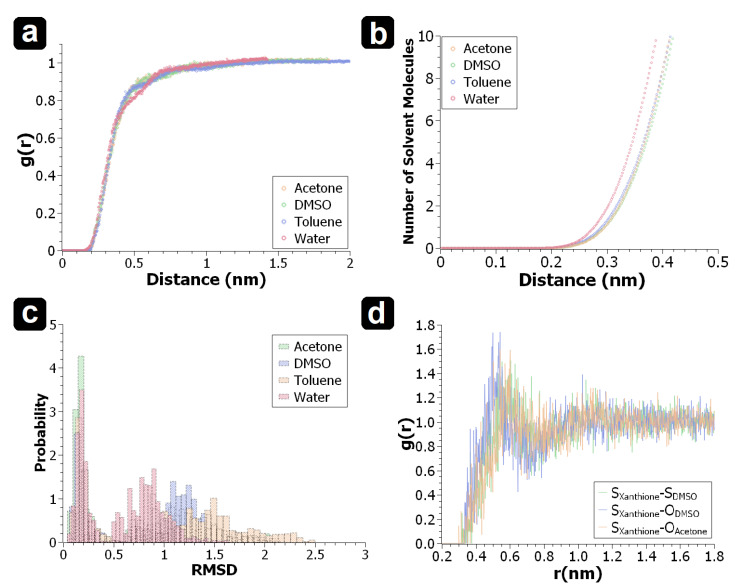
(**a**) Radial distribution function g(r) of the center of mass of solvent molecules (acetone, dimethyl sulfoxide, toluene, and water) relative to xanthione. (**b**) Nearest neighbour solvent count as a function of distance from the solute. (**c**) Root mean square deviation (RMSD) distribution of xanthione over the MD trajectories in solutions with respect to the force field optimized structure in implicit solvent. (**d**) Comparison of the radial distribution function of the sulfur atom in xanthione in acetone and dimethyl sulfoxide.

**Table 1 molecules-29-05609-t001:** Solvents, and their respective dielectric constants, paired with xanthione used in this work.

Solvent	Dielectric Constant
Toluene	2.374
1,2-Dichloroethane	10.125
Acetone	20.493
Acetonitrile	35.688
Dimethyl Sulfoxide	46.826
Formic acid	51.100
Water	78.355
Formamide	108.940

All dielectric constants reported in this work are taken as per Gaussian16 documentation [25].

## Data Availability

Electronic Appendix A contains all absolute energies (hartree) for xanthione in each solvent for implicit and explicit solvent configurations, rendered frontier molecular orbitals, and the optimised coordinates for each configurations studied in this work.

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
