# Peer review of "Simulation of Solvatochromic Phenomena in Xanthione Using Explicit Solvent Methods"

_molecules, 2024, doi:10.3390/molecules29235609_

Round 1
Reviewer 1 Report
Comments and Suggestions for Authors
Molecules
Manuscript Draft
Manuscript Number: molecules-3292532-peer-review-v1
Title: Simulation of solvatochromic phenomena in xanthione using explicit solvent methods
The manuscript presents theoretical investigation of the Xanthione, sulfated polycyclic aromatic hydrocarbon, which exhibits unique anti-Kasha properties and substantial sensitivity to its medium. To clarify long-held beliefs about specific solvent effects, they applied rigorous theoretical solvent analysis using implicit and explicit solvent media to elucidate a more complete description of solvent polarity sensitivity in xanthione using both quantum chemical and molecular dynamics techniques.
It should be noted that the introduction of explicit solvent media is not a completely straight forward process and is associated with many pitfalls that can lead to wrong conclusions.
The final result of optimization process is always dependent on the starting positions of explicit solvent molecules, so this process must be approached very carefully, bearing in mind the correct selection of the most significant interactions of the studied molecule with solvent molecules. In a solute molecule with several possible positions for interactions with the solvent, such as the Xanthione molecule, the energetically strongest interactions do not necessarily have the greatest influence on the observed property of the studied molecule at the same time. An excessive number of explicit solvent molecules can lead to their aggregation, and these interactions can be energetically significant, but most often have little or no influence on the studied molecule.
I have to conclude that the authors presented the results of a very difficult and complicated research that deserves to be published, the presented results are scientifically relevant and it will be interesting to a readers of Molecules. However, there are parts in the text that are not clear enough, so I have to recommend that the manuscript be accepted for publication in the Molecules after major changes (major revision).
In addition for authors' consideration:
The procedure for choosing the initial positions of explicit solvent molecules is not explained in any way in the text. Please give us a brief description of the procedure.
In Computational Details is stated that the two methods are used for calculation (DFT CAM-B3LYP and DFT/MRCI BHLYP) but in Results & Discussion it is not emphasized which results belong to one method and which to another method.
Constant mixing of expressions (S1 state, dark state and La) as well as (S2 state, bright state and Lb) maid confusion in text and authors should choose one of them to use in the discussion.
I think that any initial system must have the interaction of the solvent molecule with the sulfoxide group in order for the dark state to be adequately treated.
Reviewer 2 Report
Comments and Suggestions for Authors
The paper with the title “Simulation of solvatochromic phenomena in xanthione using explicit solvent methods,” submitted by A. Manian, Z. Chen, R. J. Hudson and S. P. Russo, analyzes how to simulate solvent effects on energies of the first and second singlet excited state of xanthione derivative. The authors conclude that implicit solvent effects are not sufficient and that explicit solvent molecules are necessary to properly describe these energies. They also point out that 4-6 solvent molecules are sufficient for proper modeling.
In my opinion this topic is important because if there is a good theoretical model it can save time for experimental chemists to prepare a compound with desired properties. Therefore, I think this paper should be published. However, there are some unclear things that need to be clarified and some corrections/additions that need to be made before publication. They are listed below.
- The authors used 2-6 solvent molecules. There are many places where solvent molecules can be placed. How were the input geometries created? Where were these molecules placed and why? Have the authors try different input geometries? What solvent-solute are important? Answers to these questions can help readers use the conclusions of this paper for their future research.
- I also suggest including pictures of optimized geometries in the SI.
- Coordinates should also be added to the SI.
- In addition to graphs presented in the manuscript, I think a table with the values should be added into the SI.
- Lines 106-107: the authors state that PCM model overestimates vertical exctitation energies. Where are the values with which the comparison was made?
- Lines 137-154: the authors list literature data. Values from their work should be included for comparison.
- Figures given in the SI should have an explanation.
- Figure 1: the name of the compound is not correct.
Reviewer 3 Report
Comments and Suggestions for Authors
The authors present a combined DFT/MRCI and molecular mechanics study of how solvation effects affect models of electronic excitations in xanthione. With regards to predicting vertical excitation energies, it is claimed that explicit solvation models correct some of the overestimation that is seen in implicit solvation models. This is supported by showing computed excitation energies of various implicit and explicit models at S0 and S1/S2 geometries, which show in general that the implicit model is an upper bound on the rest of the models. This is used to conclude that explicit models are strictly required to accurately model excited states, and then MD is used to show that solvation effects influence the molecular geometry and likely how it aggregates. The study reads well, and in the end it approaches the solvation aspects from quite different perspectives using molecular clusters and periodic MD simulations.
Comments:
A crucial detail is that it was not clear to me is how the explicit solvation models were generated. The authors correctly acknowledge that the choice of model geometry can have a large influence on the results, but how were the chosen geometries selected?
The text says that a 6-31 G basis set with polarization functions was used to optimize all of the cluster geometries with the CAM-B3LYP +D3 method, but it is a crucial detail to specify which polarization functions were used (e.g. 6-31G(d) vs. 6-31G(d,p) vs. etc…). A long-range corrected functional like CAM-B3LYP is known to be sensitive to basis sets, and usually larger basis sets than these are used especially where unconventional non-covalent interactions may appear (like intermolecular S…..O interactions). Also, it is clearly stated that the DFT/MRCI approach was run via single-point calculations on presupplied geometries, but how were the S1 and S2 geometries generated? (also, were the implicit-model S1 and S2 geometries optimized with a state-specific or a ground-state approach?)
It was not entirely clear to me what the rationale is behind saying X=2,4 is good enough vs. X=6. From the data in Figures 2 and 3, it seems clear that explicit models generally lie between the gas-phase and implicit solvation results, but, as is pointed out, there are clear exceptions for La at lower solvent polarities that seem to concern the choice of geometry (note that I think the discussion on line 111 should refer to La and not Lb). It looks like the results are being mostly influenced by how the solvent molecules make contact with the central ring, in the cases of the highest-polarity solvents they all try to create hydrogen bonds with the O group and then create more hydrogen bonds elsewhere, but not all of the solvents can create such hydrogen bonds. This aspect of the work (how the structure space of the solvent affects the result) should be elaborated upon as it is directly linked to the key conclusion about an explicit model being allegedly better than an implicit one. It is also stated in the main text that the “outliers” in the plots provide upper and lower bounds on all possible molecular clusters, why is that assured?
The ESI is given as an excel spreadsheet that contains data and figures, which I understand for the most part but it is far from a publishable/printable form. Beyond the formatting/readability issues, many of the legends and details are not given (for example: H-1=HOMO=1;H=HOMO;L=LUMO, isovalues of displayed isosurfaces, row labels and units, etc.). The main text directly refers to a visible contribution of the solvent to the electron density redistribution in the case of toluene, but this is not seen in the orbitals that are given in the file.
In conclusion, the study of how solvation effects influence predictions of electronic excitations in xanthione is an interesting topic, and the tactic and results that the authors present seem individually sound. However, I am not convinced that the obtained results truly support the conclusions that were drawn in the work, as very important details about how the structural spaces were explored were unclear. As such, I do not recommend that this version of the manuscript be published in Molecules.
Round 2
Reviewer 1 Report
Comments and Suggestions for Authors
Molecules
Manuscript Draft
Manuscript Number: molecules-3292532-peer-review-v2
Title: Simulation of solvatochromic phenomena in xanthione using explicit solvent methods
The manuscript presents theoretical investigation of the Xanthione, sulfated polycyclic aromatic hydrocarbon, which exhibits unique anti-Kasha properties and substantial sensitivity to its medium. To clarify long-held beliefs about specific solvent effects, they applied rigorous theoretical solvent analysis using implicit and explicit solvent media to elucidate a more complete description of solvent polarity sensitivity in xanthione using both quantum chemical and molecular dynamics techniques.
The authors responded appropriately to my previous remarks and made the necessary corrections in the text, so I have to recommend that the manuscript be accepted for publication in the Molecules.
Author Response
We thank the reviewer for their time in reevaluating our work, and are pleased to know they are happy with our corrections.